# A scalable physician-level deep learning algorithm detects universal trauma on pelvic radiographs

Chi-Tung Cheng [1,7], Yirui Wang [2,7], Huan-Wu Chen[3], Po-Meng Hsiao[4], Chun-Nan Yeh[5], Chi-Hsun Hsieh[1], Shun Miao[2], Jing Xiao[2], Chien-Hung Liao [1,6✉] & Le Lu [2]

Pelvic radiograph (PXR) is essential for detecting proximal femur and pelvis injuries in trauma patients, which is also the key component for trauma survey. None of the currently available algorithms can accurately detect all kinds of trauma-related radiographic findings on PXRs. Here, we show a universal algorithm can detect most types of trauma-related radiographic findings on PXRs. We develop a multiscale deep learning algorithm called PelviXNet trained with 5204 PXRs with weakly supervised point annotation. PelviXNet yields an area under the receiver operating characteristic curve (AUROC) of 0.973 (95% CI, 0.960–0.983) and an area under the precision-recall curve (AUPRC) of 0.963 (95% CI, 0.948–0.974) in the clinical population test set of 1888 PXRs. The accuracy, sensitivity, and specificity at the cutoff value are 0.924 (95% CI, 0.912–0.936), 0.908 (95% CI, 0.885–0.908), and 0.932 (95% CI, 0.919–0.946), respectively. PelviXNet demonstrates comparable performance with radiologists and orthopedics in detecting pelvic and hip fractures.

[1] Department of Trauma and Emergency Surgery, Chang Gung Memorial Hospital, Linkou, Chang Gung University, Taoyuan, Taiwan. [2] PAII Inc, Bethesda, MD, USA. [3] Division of Emergency and Critical Care Radiology, Department of Medical Imaging and Intervention, Chang Gung Memorial Hospital, Chang Gung University College of Medicine, Taoyuan, Taiwan. [4] New Taipei Municipal TuCheng Hospital, New Taipei city, Taiwan. [5] Department of Surgery, Chang Gung Memorial Hospital, Linkou, Chang Gung University, Taoyuan, Taiwan. [6] Center for Artificial Intelligence in Medicine, Chang Gung Memorial hospital, Linkou, Taoyuan, Taiwan. [7] These authors contributed equally: Chi-Tung Cheng, Yirui Wang. ✉email: surgymet@gmail.com

Trauma management is a race against time. A timely, accurate diagnosis with appropriate management is the key to saving lives. High-quality clinical trauma care and treatment not only rely on physician experience but also require information from various imaging modalities. Pelvic radiography is one of the diagnostic imaging modalities commonly used to evaluate blunt trauma patients. Pelvic radiographs (PXRs) cover the pelvis and the upper femur area, providing diagnostic value for orthopedic injuries in these regions, including hip fracture, pelvic fracture, hip dislocation, and other associated injuries.

Hip fractures are diagnosed on the basis of these images and are the most frequently occurring fracture type in elderly people; however, the misdiagnosis rate ranges from 4 to 9%[1], and delayed diagnosis leads to unfavorable consequences. On the other hand, pelvic ring fractures are the most life-threatening fracture, with a mortality rate exceeding 30% in unstable patients[2]. In individuals with these injuries, early diagnosis in the emergency room (ER) and early management may prevent adverse outcomes[3–5]. However, in the stressful and chaotic ER, image-based diagnosis usually relies on emergency physicians who encounter the patient first. Moreover, radiologists are not always available 24/7, especially in local hospitals or rural areas. An effective computer-aided diagnosis algorithm can prevent misdiagnoses and provides early warnings for life-threatening conditions.

Deep learning (DL) is a rapidly evolving subcategory of machine learning and is especially valuable in medical image analysis[6]. DL has been shown to be successful in performing several classification tasks, such as diagnosing skin lesions[7], analyzing retinal images[8], classifying chest radiography abnormalities[9,10], and reading neural images[11,12]. One obstacle for developing DL algorithms for medical image analysis is obtaining large-scale annotations of medical images, which is often labor-intensive and requires special expertise. To this end, numerous studies have been conducted to train deep convolutional neural networks (DCNNs) using weak labels (i.e., annotations that only indicate the presence/absence of the findings without specifying the exact location)[13–15], which can be automatically or semiautomatically obtained from medical records at a low cost. For fracture detection, several studies employed weakly supervised learning to detect fractures in local regions and demonstrated that the algorithm had a comparable accuracy to physicians[16]. Previous studies have shown that DL achieved an accuracy ranging from 90.6 to 96.1% for detecting hip fractures in various settings[17–21]. However, there could be multiple abnormal findings that appear concurrently in a PXR other than the hip fracture in a trauma patient. A clinically convincing and useful computer-aided diagnosis algorithm needs to have a universal ability to detect various pathologies on a single X-ray image. To date, few algorithms have demonstrated the ability to detect abnormalities spanning multiple categories simultaneously in an image with comparable physician-level performance. As an early version of our work, a two-stage weakly supervised method[22] was the first to achieve a hip and pelvic fracture detection performance that was comparable to that of emergency physicians and residents. However, the study still showed a significant performance gap between the two-stage model and specialized experts (i.e., orthopedic specialists and radiologists).

In this work, we hypothesize that incorporating location supervisory signals in training DL detectors can effectively improve their performance. In general computer vision, object detection tasks are usually formulated as fully supervised, given existing annotated datasets (e.g., PASCAL-VOC and MS-COCO). For common object detection methods, the localization supervision signal is typically provided in the form of bounding boxes, which specifies the object's span in the vertical and horizontal directions. However, defining the bounding box of trauma-related findings is technically difficult and practically unreliable. First, extensive clinical experience and intensive labor are required to accurately annotate bounding boxes for all fracture sites/instances on PXRs. Moreover, unlike objects in natural images, trauma-related findings on PXRs may not have clear definitions of instance and boundary, making the bounding box annotations unreliable. For instance, a pelvic fracture usually involves multiple anatomical sites with complex morphology and fragments that lead to difficulty in defining the border, extent, and number of bounding boxes. Therefore, a DL algorithm with a cost-effective and flexible annotation scheme is critical and highly desirable of universal trauma finding detection solutions for PXRs.

In this study, we developed PelviXNet, a DL detection algorithm trained using point-based annotation, an efficient, flexible, and informative labeling method to provide local information. PelviXNet can universally detect all trauma-related radiographic findings on PXRs, including hip fracture, pelvic area fracture, hip dislocation, periprosthetic fracture, and femoral shaft fracture. To the best of our knowledge, this is the first of this type of study: only detecting one specific (and easier) category of anomalies given an imaging exam, will render the DL assisted system limited and less useful in clinics. We trained the model on 5204 PXRs with point-based annotations and evaluated its performance in a clinical population test set of 1888 patients, achieving an area under the receiver operating characteristic curve (AUROC) of 0.972. Furthermore, an independent study comparing PelviXNet with 22 physicians reported accuracies of 99.5% and 94.5% in hip and pelvic fracture detection tasks, respectively, indicating that PelviXNet outperformed emergency physicians (95.0 and 90.3%) and residents (94.9 and 89.3%) and performed comparably to orthopedic specialists and radiologists (97.4 and 94.0%).

## Results

**Training set**. We retrieved 5204 PXR images that were recorded from 2008 to 2016 for algorithm development (Fig. 1). A total of 3110 images had acute trauma-related radiographic findings, resulting in 4357 annotated points (median 1, range 0–7). The demographic data are shown in Table 1. The images included 2036 (39.1%) hip fracture images, 919 (17.7%) pelvic area fracture images, 232 (4.5%) images of other abnormalities, and 2094 (40.2%) PXRs without trauma-related radiographic findings. The PelviXNet yielded an AUROC, sensitivity, and specificity of 0.997, 0.991, and 0.976, respectively, for the development dataset after training.

**Performance with a clinical scenario test set**. In 2017, a total of 1888 patients underwent PXR examinations in the emergency department because of an injury (Fig. 1). The demographic characteristics are shown in Table 1 and significantly differed between the test set and the development dataset. The distribution of the positive images represented the real incidence of the clinical situation at a trauma center, and only 32.8% of the images had acute trauma-related findings. Among all cases, 82 (4.3%) were defined as difficult cases, and 9 (0.5%) were considered misdiagnoses. Figure 2 demonstrates the universal trauma finding detection performance of PelviXNet. The AUROC of classifying the normal and abnormal conditions is 0.972 (95% confidence interval (CI): 0.960–0.983), and the accuracy, sensitivity, specificity, positive predictive value (PPV), and negative predictive value (NPV) at the defined cutoff values are 0.924 (95% CI: 0.912–0.936), 0.908 (95% CI: 0.885–0.908), 0.932 (95% CI: 0.919–0.946), 0.867 (95% CI: 0.843–0.890), and 0.954 (95% CI: 0.944–0.964), respectively. Although class imbalance is noted in

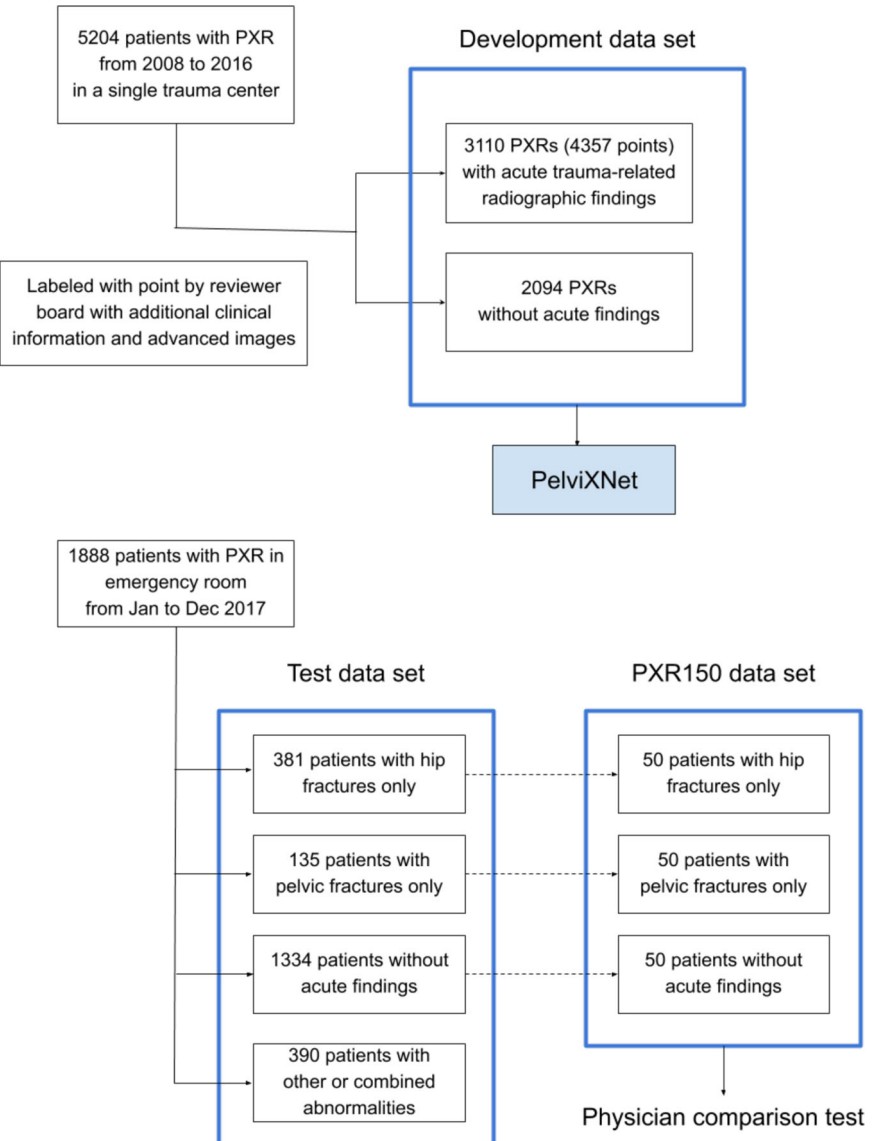

**Fig. 1 The source and distribution of each data set.** We included 5204 pelvic radiographs (PXR) from 2008 to 2016 as our development data set to develop PelviNet. Then we included PXRs from 1888 patients presented in the emergency room from January to December 2017 as our test data set. In advance, we randomized selected 150 PXRs from the test dataset to compose the PXR150 data set, which was used to perform the physician comparison test.

the dataset, the precision-recall (PR) curve still shows a promising result. The area under the PR curve (AUPRC) is 0.963 (95% CI: 0.948–0.974). Figure 3 shows examples of heatmaps for the visualization of different acute trauma-related findings.

Among all 57 false-negative images, hip ($n = 22$, 5.6%) and pelvic area fractures ($n = 23$, 15.6%) were the most common findings. Among the other, relatively rare findings, four hip dislocations, five periprosthetic fractures, and five femoral shaft fractures were not identified by PelviXNet (Table 2). Table 3 illustrates the characteristics of the acute trauma-related radiographic findings related to the pelvic area and hip fractures that were detected or missed by PelviXNet. In summary, PelviXNet has demonstrated a promising ability to detect hip fractures. Most of the missed fractures were intracapsular and difficult cases, which may require confirmation with advanced imaging modalities. Among the pelvic fractures, PelviXNet could detect most fracture sites. However, there were still four unstable pelvic fractures that were missed. A high percentage of the missed cases

were difficult cases. Surprisingly, PelviXNet detected five of the nine misdiagnosed cases.

**Evaluation of the impact of annotation size.** The size of the annotation used for training is a key factor affecting the performance of DL models. To understand how PelviXNet is affected by the annotation size, we conducted experiments to train PelviXNet with 20% ($N = 1040$), 40% ($N = 2081$), 60% ($N = 3122$), and 80% ($N = 4163$) physician annotated PXRs using the same settings and evaluated the trained models on the PXR2017 dataset. We also conducted another experiment to train the weakly supervised fracture detection model[22] for comparison using only image-level labels automatically generated from the clinical diagnosis without manual annotation from physicians. The comparisons are summarized in Table 4. We observe that the performance of PelviXNet steadily improves (i.e., AUROC from 0.933 to 0.973) as more physician-annotated PXRs (from $N = 1040$ to $N = 5204$)

are used for training. The weakly supervised model supervised by the clinical diagnosis reports an AUROC of 0.967.

**Comparison with physician performance.** The performance results of PelviXNet and physicians in analyzing the PXR150 dataset are shown in Table 5. Most physicians performed better in hip fracture detection than in pelvic fracture detection. The

performance of PelviXNet did not significantly differ from that of radiologists and orthopedic surgeons. However, PelviXNet significantly outperformed one of the ER physicians and five residents. Especially on the more difficult pelvic fracture detection task, PelviXNet's accuracy score is higher than all ER physicians and significantly better than residents, at least on par with radiologists and even specialists as orthopedic surgeons. On average, PelviXNet could detect 9%, 5.5%, and 9.7% of pelvic area fractures that were misdiagnosed by ER physicians, consulting physicians, and ER residents, respectively. Residents can potentially still benefit from the algorithm in the diagnosis of hip fractures, as PelviXNet detected 3.3% of the misdiagnosed cases.

### Discussion

A well-designed computer-aided diagnosis algorithm can reduce the occurrence of medical errors and facilitate diagnosis[23,24]. Currently, no algorithms provide a generalized and comprehensive solution for reading pelvic radiography scans in the trauma medicine domain. Although DL algorithms have previously demonstrated the ability to classify and detect abnormalities evident in radiographs, there is still a gap in the utilization of these algorithms in the clinical environment. In this study, we developed an algorithm based on a point-supervised DL method that achieved high accuracy in identifying all kinds of trauma-related radiographic findings on PXRs. PelviXNet achieved an overall accuracy of 92.4% in a real-world population dataset.

Unlike the images of human extremities, PXRs reveal a more complex anatomical structure and sometimes multiple injury sites. Most of the previous studies on PXRs have focused only on hip fractures[17–20] or osteoarthritis[15,25], which emphasize only a specific region or condition in the entire image. It is not practical to create different algorithms for each kind of abnormality that appears in a single image. Thus, we need a universal solution for a specific clinical scenario of emergency pelvic imaging examination. In our previous work[22], we tried to deal with hip fractures

| Table 1 Demographic data of development and test data set. | | | |
|---|---|---|---|
| | **No. (%) of patients** | | |
| **Variables** | **Development data set (n = 5204)** | **Test data set (n = 1888)** | **p Value** |
| Year of injury | 2008–2016 | 2017 | |
| Age, median (IQR), y | 60.00 [37.00, 78.00] | 55.00 [30.75, 75.25] | <0.001 |
| Gender, male | 2752 (52.9) | 908 (48.1) | <0.001 |
| *Mechanism of injury* | | | <0.001 |
| Motor vehicle accident | 2193 (42.1) | 893 (47.3) | |
| Fall | 2683 (51.6) | 843 (44.7) | |
| Mechanical injury | 148 (2.8) | 49 (2.6) | |
| Other mechanisms | 112 (2.2) | 42 (2.2) | |
| Unavailable | 68 (1.3) | 61 (3.2) | |
| Extremity AIS ≥ 3 | 2624 (50.4) | 517 (27.4) | <0.001 |
| AIS ≥ 16 | 1087 (20.9) | 185 (9.8) | <0.001 |
| Acute trauma finding | 3110 (59.8) | 619 (32.8) | |
| *Hip fracture* | | | |
| Intracapsular | 1014 (19.5) | 184 (9.7) | <0.001 |
| Extracapsular | 1059 (20.3) | 207 (11.0) | <0.001 |
| *Pelvic area fracture* | | | <0.001 |
| LC type | 576 (11.1) | 97 (5.1) | |
| APC type | 60 (1.2) | 7 (0.4) | |
| VS type | 25 (0.5) | 3 (0.2) | |
| Other | 259 (5.0) | 42 (2.2) | |
| Hip dislocation | 130 (2.5) | 51 (2.7) | 0.693 |
| Femoral shaft fracture | 62 (1.2) | 30 (1.6) | 0.234 |
| Periprosthetic fracture | 38 (0.7) | 29 (1.5) | 0.003 |

*IQR* interquartile range, *AIS* abbreviated injury scale, *LC* lateral compression, *APC* anterior–posterior compression, *VS* vertical shearing.

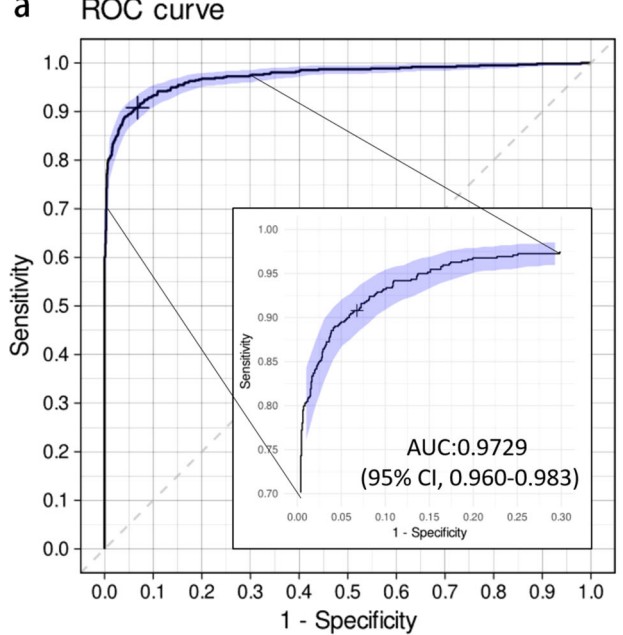

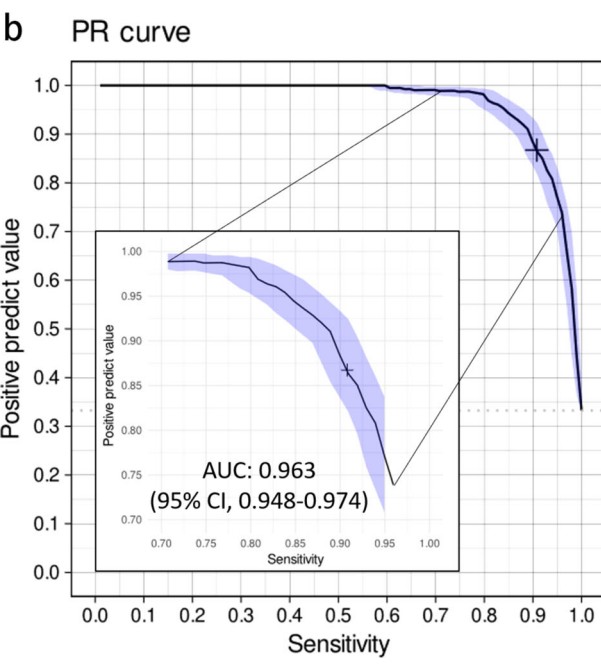

**Fig. 2 The receiver operating characteristic curve and precision-recall curve of the universal trauma finding detection algorithm. a** The receiver operating characteristic (ROC) curve of the performance of PelviXNet in the clinical scenario and the cross mark represents the performance on the probability cutoff value. **b** The precision-recall (PR) curve of the performance of PelviXNet and the cross mark represents the performance on the probability cutoff value. The 95% confidence intervals (CIs) of the ROC and PR curves were estimated using bootstrapping with 2000 replicates, which was indicated as the purple area in the panels.

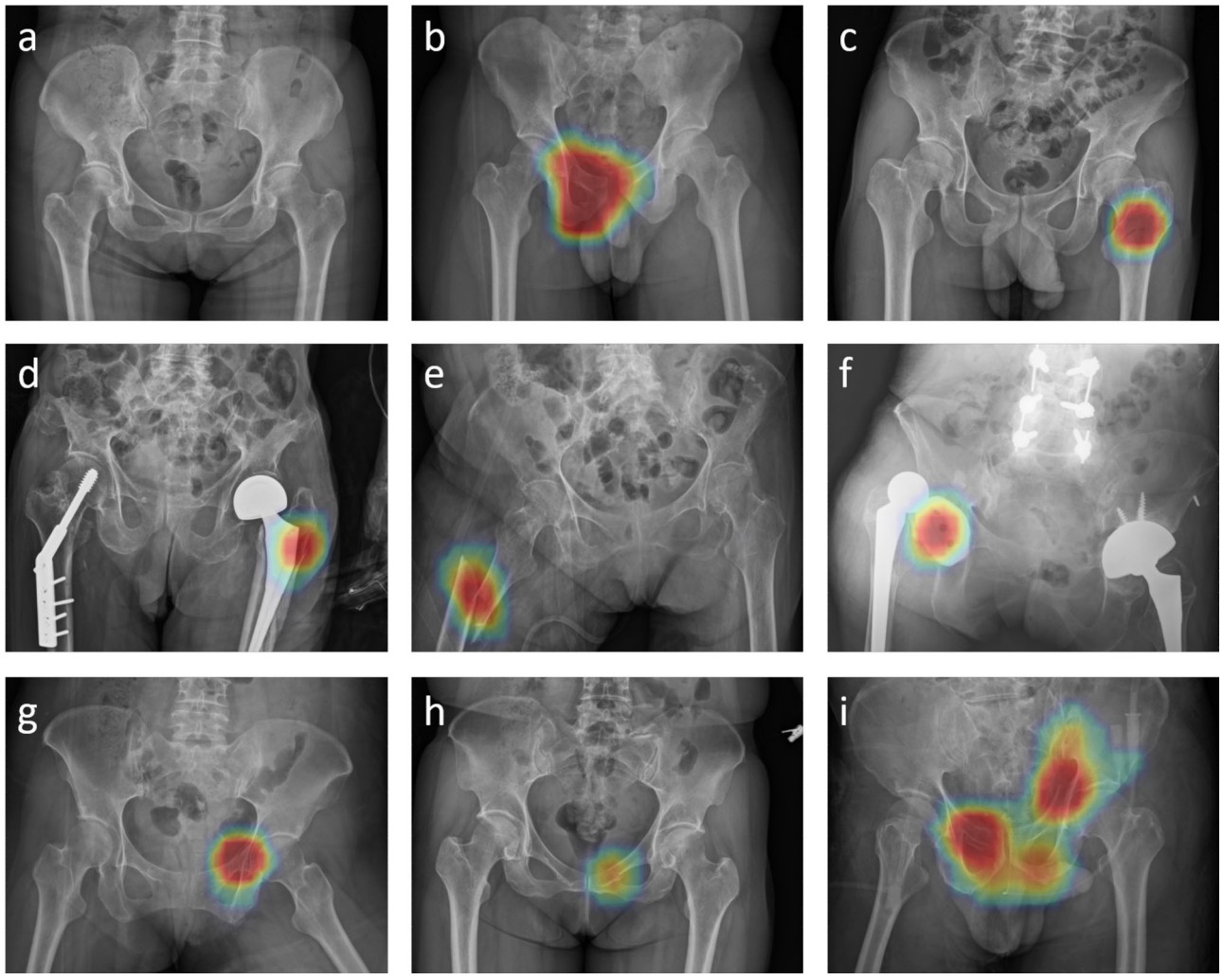

**Fig. 3 The illustration of heatmap overlaid by the algorithm on original images.** The red color represents a high probability of acute trauma finding detected. The heatmap indicates **a** no fracture. **b** Anterior–posterior compression type pelvic fracture, **c** left femoral non-displaced intertrochanteric fracture, **d** left periprosthetic fracture, **e** right femoral shaft fracture, and **f** right hip dislocation. **g** Difficult case of pelvic fracture. **h** Clinically missed pelvic fracture. **i** Multiple pelvic fractures detected simultaneously by PelviXNet, respectively. All the pelvic fracture examples except case **h** in this figure require angioembolization due to massive hemorrhage.

**Table 2 Statistics of the algorithm missed and detected trauma findings.**

|  | Algorithm missed (false-negative rate) | Algorithm detected (sensitivity) |
|---|---|---|
| Overall[a] | 57 (9.2%) | 562 (90.8%) |
| Hip fracture | 22 (5.6%) | 368 (94.4%) |
| Pelvic area fracture | 23 (15.6%) | 124 (84.4%) |
| Femoral shaft fracture | 5 (16.7%) | 25 (83.3%) |
| Hip dislocation | 4 (7.8%) | 47 (92.2%) |
| Periprosthetic fracture | 5 (17.2%) | 24 (82.8%) |

[a]Some images presented with multiple categories of injuries, the overall is counted based on the patient number.

and pelvic fractures simultaneously using an algorithm trained with image-level labels. The algorithm PelviXNet developed in this study can identify all kinds of trauma-related abnormalities and localize them correctly due to the inclusion of point supervision for the regional information in the image. The ability to identify multiple categories of abnormalities at multiple sites concurrently in an image can increase a physician's trust in the algorithm and make it more feasible to use the algorithm widely in clinical practice.

The bottleneck on developing DL models in the medical field is relatively small image numbers and the lack of labeled data. Weakly supervised methods may provide a sufficiently high baseline performance on large but a little noisy data[22]. However, some specific categories of medical images are difficult to acquire. In this study, we also evaluated the impact of the adding point annotated images on the model performance compared with a weakly supervised method using image-level information only. The model using 80% of point annotated images outperformed the weakly supervised method using all images. The result indicated adding detailed information to the model may reduce the need for training images and achieving better results.

The distribution of the dataset largely affects the test performance of the computer-aided diagnosis algorithm[26]. Although some studies have demonstrated fair levels of accuracy with a balanced dataset, the low incidence of positive findings in the clinical scenario leads to a low PPV and a large number of false-positive cases[9,27,28]. Inverse probability was also used to estimate

**Table 3 The characteristics of the algorithm identified and missed trauma findings regarding hip fracture and pelvic fracture.**

| | No. (%) of patients with hip fracture | | |
| | Algorithm missed | Algorithm identified | |
| Variables | (n = 22) | (n = 368) | p Value |
|---|---|---|---|
| Age, median (IQR), y | 71.00 [51.50, 78.50] | 79.00 [67.00, 87.00] | 0.018 |
| Gender, male | 10 (45.5) | 140 (38.0) | 0.505 |
| *Mechanism of injury* | | | 0.389 |
| Motor vehicle accident | 7 (31.8) | 61 (16.6) | |
| Fall | 15 (68.2) | 295 (80.2) | |
| Mechanical injury | 0 (0.0) | 7 (1.9) | |
| Other mechanisms | 0 (0.0) | 2 (0.5) | |
| Unavailable | 0 (0.0) | 3 (0.8) | |
| *Hip fracture* | | | |
| Intracapsular | 16 (72.7) | 168 (45.7) | 0.016 |
| Extracapsular | 6 (27.3) | 201 (54.6) | 0.015 |
| Misdiagnosed | 2 (9.1) | 1 (0.3) | 0.001 |
| Difficult case | 18 (81.8) | 34 (9.2) | <0.001 |

| | No. (%) of patients with pelvic area fracture | | |
| | Algorithm missed | Algorithm identified | |
| | (n = 23) | (n = 124) | p Value |
|---|---|---|---|
| Age, median (IQR), y | 58.00 [30.50, 77.50] | 45.00 [25.00, 66.00] | 0.11 |
| Gender, male | 7 (30.4) | 60 (48.4) | 0.171 |
| *Mechanism of injury* | | | 0.238 |
| Motor vehicle accident | 13 (56.5) | 87 (70.2) | |
| Fall | 8 (34.8) | 26 (21.0) | |
| Mechanical injury | 1 (4.3) | 5 (4.0) | |
| Other mechanisms | 0 (0.0) | 5 (4.0) | |
| Unavailable | 1 (4.3) | 1 (0.8) | |
| Hemodynamic unstable | 1 (4.3) | 5 (4.0) | 1 |
| Transarterial embolization | 1 (4.3) | 16 (12.9) | 0.475 |
| Extremity AIS ≥ 3 | 6 (26.1) | 53 (42.7) | 0.167 |
| AIS ≥ 16 | 4 (17.4) | 39 (31.5) | 0.217 |
| *Pelvic area fracture* | | | 0.804 |
| LC type | 13 (56.5) | 83 (66.9) | |
| APC type | 1 (4.3) | 6 (4.8) | |
| VS type | 0 (0.0) | 3 (2.4) | |
| Other | 9 (39.1) | 32 (25.8) | |
| Structural unstable | 4 (17.4) | 30 (24.2) | 0.597 |
| Misdiagnosed | 3 (13.0) | 3 (2.4) | 0.073 |
| Difficult case | 12 (52.2) | 10 (8.1) | <0.001 |

*IQR interquartile range, AIS abbreviated injury scale, LC lateral compression, APC anterior–posterior compression, VS vertical shearing.*

**Table 4 The comparison of models trained using different numbers of physician annotated images. The weakly supervised method uses only image-level annotations automatically produced from clinical diagnosis.**

| | Annotated images | AUROC | AUPRC |
|---|---|---|---|
| Supervised | N = 1040 (20%) | 0.933 | 0.921 |
| | N = 2081 (40%) | 0.959 | 0.949 |
| | N = 3122 (60%) | 0.961 | 0.952 |
| | N = 4163 (80%) | 0.969 | 0.963 |
| | N = 5204 (100%) | 0.973 | 0.963 |
| Weakly supervised | Image-level only | 0.967 | 0.957 |

*AUROC area under the receiver operating characteristic curve, AUPRC area under the precision-recall curve.*

have a performance comparable to that of physicians in detecting proximal humerus fractures[30], wrist fractures[23], and hip fractures[18,20] on radiographs. In our study, PelviXNet performed similarly with consulting physicians, such as radiologists and orthopedic surgeons. Some hip fractures and pelvic fractures that ER physicians missed could be detected with PelviXNet using the classification confidence scores and class activation heatmaps to indicate and localize findings. This means that frontline physicians can receive real-time recommendations from the algorithm when they are treating multiple trauma patients, as misdiagnoses can occur in a chaotic ER[31]. Regarding pelvic fractures, most structurally and hemodynamically unstable pelvic fractures can be detected by our algorithm, which is especially useful in institutes lacking consulting specialists or experienced medical staff[32].

Trauma is time-sensitive, and early treatment relies on an accurate and rapid diagnosis[33]. The task of detecting all kinds of trauma-related findings on PXRs using an algorithm is one critical component of an automation-assisted trauma diagnosis workflow. Furthermore, computer-aided diagnosis systems can be applied in institutes lacking specialists or can even be used in prehospital settings[34]. Evidence shows that early treatment for hip and pelvic fractures can reduce morbidity and mortality[3–5,35–37]. Rapid diagnosis is the key to achieving early damage control and providing definite treatment. Collaboration between the algorithm and physicians can improve the quality of trauma care.

**Limitations.** In addition to our algorithm achieving a physician-level performance in detecting hip fractures, this is the first study to create an algorithm that can detect pelvic fractures to the best of our knowledge. Nevertheless, there were still some limitations of this algorithm. The primary reason for the main limitation is the paucity of the training data. The DL-based algorithm is a data-driven method that relies on considerable data to solve problems[38]. Insufficient training data, such as hip dislocation, will decrease algorithm performance. Fortunately, the conditions analyzed in this study are rarely missed by physicians[39,40], so the benefits of detecting these abnormalities are marginal compared to diagnoses made with a computer-aided diagnosis algorithm. Another limitation is that this was a retrospective image review study limited to a single institute. The population and images collected may be biased by this setting and might not directly apply to other institutes with different population distributions. In this study, we randomly selected those images based on the clinical diagnosis before the whole analysis started. Therefore, there is probably selective bias. Furthermore, we evaluated the performance of physicians and the algorithm in terms of detecting the main category of the disease of interest on images. However, in real-world scenarios, physicians make diagnoses on

the algorithm behavior with the population distribution[9]. In our study, we collected all PXR images taken in 2017 in the ER by consecutive sampling. Although the overall number of positive cases accounted for only 32.7% of the cases, considering for a trauma center where relatively severe conditioned patients were sent in, our algorithm still achieved a PPV of 86% while maintaining a sensitivity of 90.8%.

Another consideration for DL algorithms is that they need to have a performance comparable to that of physicians to yield clinical benefits[29]. Previous studies have shown that algorithms

**Table 5 The comparison of performance between physicians and the algorithm regarding hip fracture and pelvic fracture.**

| | Hip fracture | | | | Pelvic fracture | | | |
|---|---|---|---|---|---|---|---|---|
| | Acc. | Sen. | Spe. | Misdiagnosis detected | Acc. | Sen. | Spe. | Misdiagnosis detected |
| Our algorithm | 0.995 | 1.00 | 0.99 | | 0.945 | 0.92 | 0.97 | |
| ER physician 1[a] | 0.915 | 0.94 | 0.89 | 0 (0%) | 0.925 | 0.90 | 0.95 | 2 (4%) |
| ER physician 2 | 0.955 | 0.98 | 0.93 | 1 (2%) | 0.865 | 0.78 | 0.95 | 8 (16%) |
| ER physician 3 | 0.970 | 1.00 | 0.94 | 0 (0%) | 0.915 | 0.86 | 0.97 | 4 (8%) |
| ER physician 4 | 0.960 | 0.98 | 0.94 | 0 (0%) | 0.905 | 0.86 | 0.95 | 4 (8%) |
| Mean | 0.950 | 0.975 | 0.925 | 0.25 (0.5%) | 0.903 | 0.850 | 0.955 | 4.5 (9.0%) |
| Radiologist 1 | 0.985 | 1.00 | 0.97 | 0 (0%) | 0.940 | 0.88 | 1.00 | 3 (6%) |
| Radiologist 2 | 0.970 | 0.98 | 0.96 | 1 (2%) | 0.925 | 0.86 | 0.99 | 4 (8%) |
| Orthopedics 1 | 0.960 | 1.00 | 0.92 | 0 (0%) | 0.970 | 0.94 | 1.00 | 0 (0%) |
| Orthopedics 2 | 0.980 | 1.00 | 0.96 | 0 (0%) | 0.925 | 0.86 | 0.99 | 4 (8%) |
| Mean | 0.974 | 0.995 | 0.953 | 0.25 (0.5%) | 0.940 | 0.885 | 0.995 | 2.75 (5.5%) |
| Resident 1[a] | 0.955 | 0.98 | 0.93 | 0 (0%) | 0.800 | 0.62 | 0.98 | 16 (32%) |
| Resident 2 | 0.970 | 1.00 | 0.94 | 0 (0%) | 0.880 | 0.86 | 0.90 | 4 (8%) |
| Resident 3 | 0.995 | 1.00 | 0.99 | 0 (0%) | 0.920 | 0.86 | 0.98 | 4 (8%) |
| Resident 4[a] | 0.890 | 0.94 | 0.84 | 1 (2%) | 0.810 | 0.68 | 0.94 | 13 (26%) |
| Resident 5 | 0.980 | 0.96 | 1.00 | 2 (4%) | 0.910 | 0.86 | 0.96 | 4 (8%) |
| Resident 6[a] | 0.930 | 0.94 | 0.92 | 0 (0%) | 0.825 | 0.90 | 0.75 | 2 (4%) |
| Resident 7 | 0.930 | 0.92 | 0.94 | 3 (6%) | 0.935 | 0.92 | 0.95 | 1 (2%) |
| Resident 8 | 0.940 | 0.92 | 0.96 | 1 (2%) | 0.880 | 0.86 | 0.90 | 4 (8%) |
| Resident 9[a] | 0.945 | 1.00 | 0.89 | 0 (0%) | 0.920 | 0.90 | 0.94 | 2 (4%) |
| Resident 10 | 0.965 | 0.96 | 0.97 | 1 (2%) | 0.910 | 0.86 | 0.96 | 5 (10%) |
| Resident 11 | 0.935 | 0.90 | 0.97 | 5 (10%) | 0.940 | 0.90 | 0.98 | 2 (4%) |
| Resident 12[a] | 0.885 | 0.78 | 0.99 | 10 (20%) | 0.875 | 0.78 | 0.97 | 8 (16%) |
| Resident 13 | 0.975 | 1.00 | 0.95 | 0 (0%) | 0.945 | 0.90 | 0.99 | 2 (4%) |
| Resident 14 | 0.990 | 1.00 | 0.98 | 0 (0%) | 0.950 | 0.92 | 0.98 | 1 (2%) |
| Mean | 0.949 | 0.950 | 0.948 | 1.64 (3.3%) | 0.893 | 0.844 | 0.941 | 4.86 (9.7%) |

[a]Significant difference between a physician and the algorithm on McNemar's test. Abbreviations: *Acc* accuracy, *Sen* sensitivity, *Spe* specificity.

the basis of not only radiographic findings but also clinical information, such as patient histories and physical examination findings. The actual benefit of this algorithm should be evaluated in a prospective randomized setting in a clinical environment.

This study demonstrated that a universal trauma-related detection algorithm for PXRs can be trained scalably using point-based weakly supervised annotations and achieve a suitable performance both in a clinical scenario distribution dataset and balanced dataset. This is the first algorithm to detect pelvic and hip fractures concurrently, which can prevent the misdiagnosis of these injuries in a clinical setting. Future prospective studies are required to validate whether applying PelviXNet as a computer-aided diagnostic system in a clinical environment leads to more precise diagnoses and facilitates trauma patient management.

## Methods

**Data sources**. The development dataset was established by retrospectively reviewing the data in the trauma registry of the Chang Gung Memorial Hospital in Taiwan recorded from May 2008 to December 2016. The demographic and trauma-related data, including age, sex, date of injury, mechanism of injury, vital signs upon arrival, final diagnosis, abbreviated injury scale score, injury severity score, and outcomes, were recorded. The first pelvic anteroposterior radiograph taken after the patient's arrival was acquired from the picture archiving and communication system repository. The images were deidentified and converted to portable network graphics format for further processing. An image review board consisting of a radiologist, a trauma surgeon, and an orthopedic surgeon who had 15 years, seven years, and three years of experience, respectively, was responsible for identifying the trauma-related findings. The best available information was provided, including the original images, radiologist reports, clinical diagnoses, surgical reports, and advanced imaging modality findings, if available. The reviewers were asked to annotate the images by marking the center of the trauma-related radiographic findings, such as hip fractures, pelvic area fractures, hip dislocations, periprosthetic fractures, and femoral shaft fractures. Pelvic ring fractures, acetabular fractures, sacrum fractures, and ilium fractures were all defined as pelvic area fractures. Multiple marks were made if more than one abnormal site was identified. The three reviewers labeled the images separately. When the

inconsistency happens, one of the reviewers will review the clinical information and additional image exams of the patient. He will also discuss with the other two labels to make a final decision of the label.

To evaluate the algorithm performance with distribution data from a real-world population, an independent clinical scenario test set was retrieved from the ER registry data recorded from January 2017 to December 2017 and named the PXR2017 test set. All images acquired in the ER as indicated by trauma were collected. The test dataset did not overlap with the development dataset. The clinical information retrieved and the manner in which the images were annotated were the same as those used for the development dataset. Additionally, difficult cases were assessed by the review board if the finding was controversial on the initial PXR image or considered visually insignificant. Misdiagnosed cases were also identified when the review board considered positive findings to be evident on the image, but no medical records documented the disease. This study was approved by the Institutional Review Board of Chang Gung Memorial hospital, with the identification number CGMHIRB 201702059B0. The consent of data collection was waived by the IRB. All of the participating physicians were well-informed about the design of the study and algorithm, and the written consents were collected.

**Algorithm design**. PelviXNet was designed with a combination of DenseNets, point supervision, and FPNs (illustrated in Fig. 4). In brief, only the development dataset was used to develop the algorithm. The images were resized to 1024 × 1024 pixels as inputs in the DL algorithm. The output is a 32 × 32 binary classification probability heatmap, demonstrating the possibility of trauma-related radiographic findings. The heatmap could further be overlaid on the original image to visualize the results produced by the algorithm. A point-based supervision technique was used to extract the local information from positive image regions during the training process. The image-level prediction was generated on the basis of the maximum value of the heatmap. After the training images were repeatedly inputted into the algorithm, it adjusted the calculated weight inside the network, and the performance converged, demonstrating that the algorithm had the learning ability to universally detect trauma-related radiographic findings. During the inference stage, five models with five augmented images each were ensembled to generate the final prediction of the image (Fig. 5).

**Image preprocessing**. In a preprocessing step, all PXRs were padded to square shapes using zeros and resized to 1024 × 1024. The preprocessing step standardizes the input size for CNNs and optimizes the graphics processing unit (GPU) memory footprint. A previous study demonstrated that appropriate data augmentation can

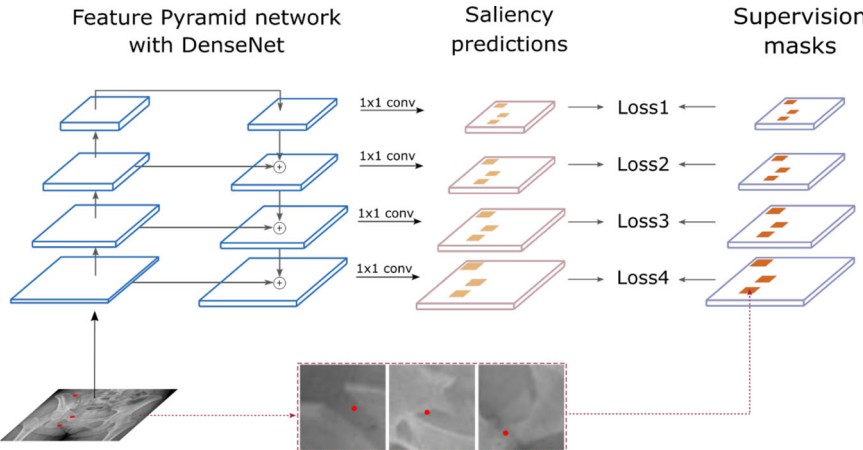

**Fig. 4 The illustration of bone fracture classification and localization system.** The proposed bone fracture classification and localization system trained with point-based supervision signals. The network uses PXR images as input and extracts different levels of feature abstractions through a bottom-up pathway, which is further fused by a top-down path.

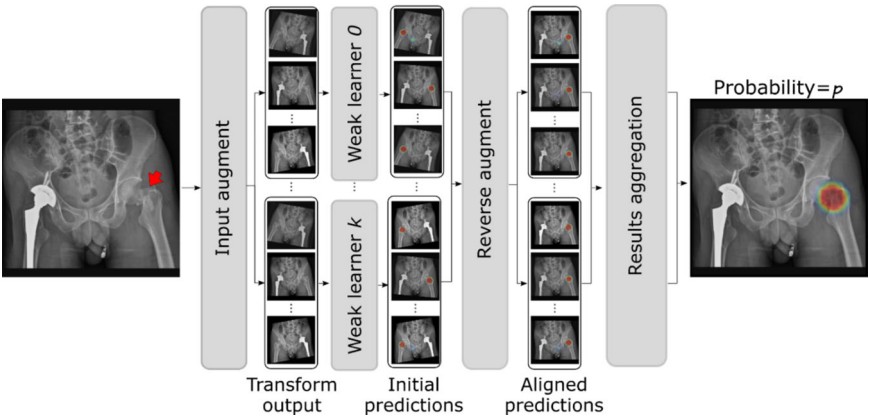

**Fig. 5 The Overviw of the ensemble method for inference in testing.** During the inference stage, once the image was input, five models with five augmented images each were ensembled to generate the final prediction of the image.

effectively improve the model's generalization ability and robustness in unseen domains[41]. During training, we further augmented the preprocessed PXR with the following operations: (1) random horizontal and vertical translations with the offset within [−100, 100] pixels in both directions; (2) random rescaling with a ratio within [0.9, 1.1]; (3) random horizontal flipping; (4) random rotation with a rotation within [−15°, 15°]; and (5) brightness and contrast jittering with a magnitude ratio within [0.75, 1.25].

**Point-base annotation.** Object detection techniques have been extensively studied by the computer vision society in recent decades. The most common form of annotation is the bounding box, which is placed on each instance of the target object[42–44]. However, the nature of pelvic fractures is inherently different from that of objects in regular object detection tasks (vehicle, animal, tree, etc.). Due to the complex morphology of pelvic fractures, it is often difficult to define the border, extent, and number of bounding boxes. In addition, bounding box annotation is labor-intensive, making it less economically preferable in large-scale medical image annotation. To bridge this gap, in this study, we proposed a DL detection algorithm trained using point-based annotation, an efficient yet informative labeling method to provide findings' localization information conveniently. The pin-pointing annotation rule is flexible and naturally fits for complex scenarios: we asked the annotators/physicians to place points on visible fracture sites. For complex scenarios where the instance of fractures cannot be clearly defined, the annotators decided to place one or multiple points at their own discretion. The training mechanism of PelviXNet was designed to accept the thus formed point-based annotations by effectively handling annotation variations caused by ambiguous or challenging scenarios.

To produce the supervision signals for training the CNN from point-based annotations, the points were converted to make the image masks of size 1024 × 1024 per X-ray image, with areas indicating the locations of fracture sites. For each

PXR, the mask was generated with "ones" for pixels within an empirically selected radius of $s = 75$ pixels from any annotation points (i.e., each annotation point was represented by a disk in the mask) and "zeros" elsewhere. Since the exact scale and shape of the pelvic radiographic findings are unknown solely based on the annotation point, the generated masks provided noisy yet informative pixel-level supervision signals for training the CNN detector. Previous studies have demonstrated that DL is robust to label noises[45].

**Training the DL model.** Due to the intrinsically different types and natures of pelvic radiographic findings (pelvic ring fractures, acetabular fractures, sacrum fractures, ilium fractures, etc.), their radiographic patterns can vary considerably in scale. To model the radiographic patterns on different scales, we employed DenseNet-169 a feature pyramid network (FPN)[44] as the backbone feedforward neural network (illustrated in Fig. 4). FPNs have been widely adopted in object detection techniques for natural images to cope with the size differences of objects caused by image resolution and perspective effects[43]. The FPN outputs multiple levels of feature maps with different spatial resolutions, which can be used to capture objects of different sizes. In anchor-based object detection methods, object annotations were assigned to different pyramid levels according to their size measured by the bounding box. For fracture detection on PXR with point-based annotation, the size of the fracture was inherently ambiguous (because of both perspectives of the complexity of the fractures and the form of annotation). Therefore, we assigned every fracture annotation point to all pyramid levels in the FPN to encourage the network to recognize the fractures at different spatial resolutions.

From PelviXNet, the backbone DenseNet-169 has a bottom-up path that encoded the input PXR using four sequential blocks, each downsampling the image size by 2. The FPN added a top-down path that upsamples and fused the feature maps produced in the bottom-up path, resulting in four feature maps {F1, F2, F3,

F4} with spatial resolutions of $32 \times 32$, $64 \times 64$, $128 \times 128$, and $256 \times 256$, respectively. The feature maps were then processed by a $1 \times 1$ convolution to produce dense saliency maps of positive radiographic findings, denoted as {P1, P2, P3, P4}. During training, saliency maps at all four levels were compared with the supervision masks to calculate training loss. Specifically, the supervision mask was downsampled to the spatial resolution of {P1, P2, P3, P4}, denoted as {M1, M2, M3, M4}. The neural network was trained using binary cross-entropy (BCE) loss calculated pixel-wise on the four saliency maps and supervision masks:

$$\mathcal{L} = \sum_{k} \frac{1}{\Omega_k} \sum_{i,j} -[M_k(i,j) \cdot \log \sigma(P_k(i,j)) + (1 - M_k(i,j)) \cdot \log \sigma(1 - P_k(i,j))] \dots$$
(1)

where $(i, j)$ denotes the index of the pixel, k denotes the index of the output saliency map, $\Omega_i$ denotes the number of pixels in the output, and $\sigma(\cdot)$ denotes the sigmoid activation function. During inference, only the saliency map $P_4$ is produced as an output.

We performed fivefold cross-validation in training and model ensembling inference in testing. Under fivefold cross-validation, the development set was randomly and evenly split into fivefold per each iteration, fourfold were selected for training and the rest for validation sets. PelviXNet was trained on each training set and validated using the corresponding validation set.

**Implementation details**. DL models were developed on a workstation with a single Intel Xeon E5-2650 v4 CPU @ 2.2 GHz, 128 GB RAM, 4 NVIDIA TITAN V GPUs. The operating system was Ubuntu 18.04 LTS. All code used in this study was written in Python v3.6, and DL models were implemented by using PyTorch v1.3. Image preprocessing was performed using the Python Imaging Library (Python Imaging Library). We used ImageNet pretrained weights to initialize the backbone network DenseNet-169[46]. The Adam optimizer[47] was used to train the model for 100 epochs with a batch size of 8 and a starting learning rate of 1e−5. The trained model was evaluated on the validation set after every training epoch, and the one with the highest validation AUROC was selected as the best model.

**Inference with ensemble learning**. Previous studies have demonstrated that the generalization ability of an ensemble of multiple learners can be significantly better than that of single learners[48,49]. In this work, by conducting fivefold cross-vali-dation, the five best models were selected based on their validation AUROCs and could be considered weak learners under the ensemble learning setup. In this work, we adopt a bagging strategy[50] to combine weak learners into an ensemble model to reduce the variance of weak learners. Figure 5 shows our ensemble learning system at the inference time. For each weak learner, the input PXR was preprocessed to $1024 \times 1024$ and augmented five times using predefined magnitude combinations of horizontal flipping, rotation, and contrast. The model inference was then per-formed individually on the five augmented PXRs. This step aimed to produce inference results of the input PXR under different appearance perturbations, which offer diversified results that can increase the robustness of ensemble learning. With the five best models each inferred from five augmentations, in total, 25 saliency maps were produced, denoted as $P^n$. The final image-level probability was calcu-lated by an ensemble of the average of $P^n$, written as:

$$p = \frac{1}{25} \sum_{n=1}^{25} \max_{i,j} \sigma(P^n(i,j)), \dots$$
(2)

The final localization map for radiographic findings was produced by the ensemble of the 25 saliency maps. Specifically, an inverse transform of the augmentation was applied for each saliency map to spatially align them with the original input PXR. Then, a pixel-wise average of the saliency maps was calculated to produce the final output.

*Independent clinical scenario test evaluation*. After the algorithm was developed, it was applied to the PXR2017 dataset to evaluate its universal trauma-related per-formance in detecting radiographic findings in the patient data from a real-world clinical population. The cutoff probability of trauma-related findings being present was defined by the review board to balance sensitivity and specificity for the clinical application. The ROC curve and PR curve were calculated. The sensitivity, speci-ficity, PPV, and NPV at the cutoff value were evaluated.

**Comparison with physician performance**. In the ER, the most important and frequent clinical application of PXRs is to detect hip fractures and pelvic area fractures. From the PXR2017 dataset, we randomly selected 50 hip fracture images, 50 pelvic area fracture images, and 50 normal images and included them in the PXR150 test set[51] to compare the diagnostic performances of the algorithm and physicians. We recruited four ER physicians, two radiologists, two orthopedic surgeons, and 14 residents who worked in the ER and evaluated their diagnostic performances. The participants were asked to perform a web-based test to classify the type of fractures that were presented in a randomly ordered PXR150 test set. PelviXNet[52] was also applied to this dataset. The prediction heatmap was reviewed to confirm the classifications of the fractures. The results of the physicians were compared with those of PelviXNet. The acute trauma-related findings that

PelviXNet detected but that the physicians missed were considered potential misdiagnosis cases that could have been prevented.

**Statistical analysis**. All statistical analyses were carried out using R 3.6.3 with the packages "pROC", "tableone", "caret", and "ggplot2". The continuous variables were analyzed with the Kruskal–Wallis rank-sum test, and the categorical variables were compared with the chi-square test and Fisher's exact test. The AUROC and AUPRC were calculated. The 95% CIs of the ROC and PR curves were estimated using bootstrapping with 2000 replicates. Youden's $J$ statistic was used to determine the performance of a given cutoff value. The performance of PelviXNet and the physicians with the PXR 150 test set was compared with McNemar's test. The balanced accuracy and class-specific sensitivity and specificity were demonstrated for the three-class classification problem in the PXR150 test. A $p < 0.05$ indicated statistical significance.

**Reporting summary**. Further information on research design is available in the Nature Research Reporting Summary linked to this article.

## Data availability
The imaging data are not publicly available due to the restricted permissions of the current study and the policy of the institute. The test imaging dataset for evaluates the performance was available on https://doi.org/10.34747/f06m-m978 for data validation use and academic purpose only.

## Code availability
The code used to evaluate the model performance is publicly available on https://doi.org/10.34747/3haq-pv57. Usage of this code is for academic purposes only and the operation packages were accessed upon request.

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

## Acknowledgements
The authors thank CMRPG3H0971, CMRPG3J631, NCRPG3J0012 (MOST-108-2622-B-182A-002), and CIRPG3H0021 for supporting the development of the system. The authors received funding from Chang Gung Memorial hospital research grant CMRPG3H0971, CMRPG3J631, CIRPG3H0021 to support this work.

## Author contributions
All the authors made substantial contributions to this work. C.T. Cheng and C.H. Liao present the conception of the work; L. Lu and C.N. Yeh design the study; H.W. Chen, P. M. Hsiao, and C.H. Liao acquired the data and labeling the images for further analysis. J. Xiao, L Lu, Y Wang, and S Miao analyzed the data and develop the deep learning algorithm.; Y Wang, and S Miao created the new software for labeling image used in the work; CT Cheng, CN Yeh, and Y Wang have drafted the work; CH Liao and CH Hsieh substantively revised it

## Competing interests
The authors declare no competing interests.
