## [Peer Review File · Nature Communications]

Reviewers' Comments:

Reviewer #1:

Remarks to the Author:

The paper presents a novel tool for the automatic evaluation of pelvis radiographs, aimed at the detection of hip and pelvic fractures. Results showed a performance on par with skilled human observers, and superior to less experienced evaluators such as ER physicians and residents. The assessment of the results is very rigorous. The work is undoubtedly valuable and merits publication. I have one technical observation:

- in the Introduction, the Authors state (I agree with them) that exploiting readily-available data from medical records through weakly supervised methods would be advantageous, since the major bottleneck with deep learning is the availability of large annotated datasets. However, the present method is still based on a fully supervised approach (even if no bounding boxes are required, and annotation is arguably faster and less tedious). Therefore, the major bottleneck is still there, and I think that this issue should be better discussed in the Discussion.

Reviewer #2:

Remarks to the Author:

Nature Communications Review

Manuscript ID#: NCOMMS-20-31764

Manuscript Title: A Physician-Level Scalable Deep Learning Algorithm of Universal Trauma Finding Detection of Pelvic Radiographs

General Comments:

The investigators have performed a study, the purpose of which was to utilize a deep learning algorithm for detection of abnormalities on pelvic radiographs. The patient morbidity associated with pelvic fractures and dislocations is substantial, and any DL algorithm assessment utilized to improve accuracy would be welcome. The motivation of automating pathology detection in a specific clinical setting (ED, for example), at a level near radiologist performance is well defined and addresses a clinically relevant condition. Concerns are raised, however, regarding methodology in the presented study.

While the general ML-methods, including model architecture, data augmentation, train/test split, and point-based innovation are admirable, important signals of performance are missing, as described below, and a lack of a strong claim on what their model performs well on (e.g., hip/pelvic fractures only or for all considered pathologies), seem to combine to potentially cherry-pick good results.

Additional concerns are raised such as a clear explanation of their ground truthing scheme and operating point determination of classification. Of note, the model missed four hip dislocations (lines 158-159), which the authors dismiss as a "rare finding." Indeed, this underscores the difficulty and limitation of using a single AP projection, which assess lateral translation, without a frog lateral or additional lateral projection to assess the AP position of the femur relative to the acetabulum. Missed hip dislocations are potentially devastating, and this strongly limits the clinical application of the algorithm.

Specific Comments:

1. Abstract Lines 42-44

The investigators indicated that the algorithm could detect "all major types of trauma-related radiographic findings," but as noted above, missed four hip dislocations (line 158).

2. Introduction Lines 121-123 –

A positive point is that the investigators built off of previous work by training with weak localization labels and getting more data, using an innovative labeling approach for fast and easy localization labels ("point-based annotation").

3. Methods

Some concern is raised regarding the discordance between the size of the test and training set (5204 PXR's versus PRX150 test set (50 hip fractures, 50 pelvic area images, and 50 normal images)). The authors indicate that one of their limitations was their relatively small training set. While it is true that more data would likely improve performance, there is no reason to believe that this is, in fact, the case (e.g., showing model performance as a function of size of the data set). Ensembling strategy both at training time (through cross fold validation) and a test time (through data augmentation) are both sound ways to improve model performance, although at a cost of inference time speed when deploying the algorithm in practice.

4. Methodology –

It is uncertain how the investigators aggregate labels from their three labelers, or how they chose an operating point to calculate sensitivity, specificity, and accuracy. Were dislocations definitely included in the test set?

5. Results Line 148 –

How was the 0.972 AUROC calculated; Normal vs. Abnormal (across all conditions)? Mean of AUC across conditions?

6. Table 1 –

The Table 1 breakdown of demographic data shows concern of overlap of “mechanism of injury” between development and test sets, but this reviewer is uncertain if this is a typographical error.

7. Results –

No comparison is provided of the results to previous work with only image-based labels. It seems like there are more training data now as well. How do we know that the localized labelling actually helped improve results?

In Table 4, model sensitivity and hip fractures in the randomly detected PXR150 is 100%, whereas in Table 2, it is only 99.4%. This doesn't seem like a random sample.

8. Discussion, 199 –

The authors reference their previous work as an abstract. Has this been published?

9. Figure 1 –

In Figure 1, the development data set included 5204 patients with a pelvic x-ray from a trauma center. In the annotation to the left in Figure 1, it is indicated this is labeled with “point by reviewer board with additional clinical information and advanced images.” This might potentially introduce additional bias into radiographic interpretation, relying on advanced imaging or clinical information rather than establishment of ground truth by radiographic assessment alone; i.e., “the labeler should see exactly what the model sees”.

Response to referees

Reviewer #1 :

The paper presents a novel tool for the automatic evaluation of pelvis radiographs, aimed at the detection of hip and pelvic fractures. Results showed a performance on par with skilled human observers, and superior to less experienced evaluators such as ER physicians and residents. The assessment of the results is very rigorous. The work is undoubtedly valuable and merits publication. I have one technical observation:

Comment: in the Introduction, the Authors state (I agree with them) that exploiting readily-available data from medical records through weakly supervised methods would be advantageous, since the major bottleneck with deep learning is the availability of large annotated datasets. However, the present method is still based on a fully supervised approach (even if no bounding boxes are required, and annotation is arguably faster and less tedious). Therefore, the major bottleneck is still there, and I think that this issue should be better discussed in the Discussion.

Response: Thanks for your constructive suggestion. We added an experiment to evaluate the impact of the amount of point annotated images to the model. Using 80% of point annotated images, the model outperformed the weakly supervised method using all images with image-level information only. As you suggested in the discussion, we also added a paragraph to describe the significant bottleneck of developing medical image deep learning models.

Line 169 -180: The size of the annotation used for training is a key factor affecting the performance of deep learning models. To understand how PelviXNet is affected by the annotation size, we conducted experiments to train PelviXNet with 20% (N=1040), 40% (N=2081), 60% (N=3122) and 80% (N=4163) physician annotated PXR's using the same settings and evaluate the trained models on the PXR2017 dataset. We also conducted another experiment to train the weakly supervised fracture detection model²² for comparison using

only image-level labels automatically generated from the clinical diagnosis without manual annotation from physicians. The comparisons are summarized in Table 6. We observe that the performance of PelviXNet steadily improves (i.e., AUROC from 0.933 to 0.973) as more physician annotated PXR (from N=1040 to N=5204) are used for training. The weakly supervised model supervised by the clinical diagnosis reports an AUROC of 0.967.

Line220-229: The bottleneck on developing deep learning models in the medical field is relatively small image numbers and the lack of labeled data. Weakly supervised methods may provide a sufficiently high baseline performance on large but a little noisy data. However, some specific categories of medical images are difficult to acquire. In this study, we also evaluated the impact of the adding point annotated images on the model performance compared with a weakly supervised method using image-level information only. The model using 80% of point annotated images outperformed the weakly supervised method using all images. The result indicated adding detailed information to the model may reduce the need for training images and achieving better results.

Reviewer #2 :

General Comments:

The investigators have performed a study, the purpose of which was to utilize a deep learning algorithm for detection of abnormalities on pelvic radiographs. The patient morbidity associated with pelvic fractures and dislocations is substantial, and any DL algorithm assessment utilized to improve accuracy would be welcome. The motivation of automating pathology detection in a specific clinical setting (ED, for example), at a level near radiologist performance is well defined and addresses a clinically relevant condition. Concerns are raised, however, regarding methodology in the presented study.

While the general ML-methods, including model architecture, data augmentation, train/test split, and point-based innovation are admirable, important signals of performance are missing, as described below, and a lack of a strong claim on what their model performs well on (e.g., hip/pelvic fractures only or for all considered pathologies), seem to combine to potentially cherry-pick good results.

Comment: Additional concerns are raised such as a clear explanation of their ground truthing scheme and operating point determination of classification. Of note, the model missed four hip dislocations (lines 158-159), which the authors dismiss as a “rare finding.” Indeed, this underscores the difficulty and limitation of using a single AP projection, which assesses lateral translation, without a frog lateral or additional lateral projection to assess the AP position of the femur relative to the acetabulum. Missed hip dislocations are potentially devastating, and this strongly limits the clinical application of the algorithm.

Response: Thank you for your critical comment about the deficit of the current algorithm.

In our training set, the number of hip dislocations is relatively low (130 /5204). Therefore, the performance of the algorithm to detect dislocation might not be acceptable as other kinds of trauma. However, in the testing set, the sensitivity of dislocation detection is still 92.2% (47/51), so we describe our PelviXNet could detect “most” major types of trauma-related radiographic findings. However, as your words, the prognosis of missed diagnosis of hip dislocation is dismal. Fortunately, hip dislocation is rarely missed by clinical physicians. We described and discussed this point in the limitation part to make the audience understand the limitation of the algorithm well. Thank you again.

Line 267-273 : The primary reason for the main limitation is the paucity of the training data. The DL-based algorithm is a data-driven method that relies on considerable data to solve problems.³⁸ Insufficient training data, such as hip dislocation, will decrease algorithm performance. Fortunately, the conditions analyzed in this study are rarely missed by physicians,^{39,40} so the benefits of

detecting these abnormalities are marginal compared to diagnoses made with a computer-aided diagnosis algorithm.

Specific Comments:

Comment: 1. Abstract Lines 42-44

The investigators indicated that the algorithm could detect “all major types of trauma-related radiographic findings,” but as noted above, missed four hip dislocations (line 158).

Response: As mentioned above, the performance to detect hip dislocation seems not as good as other types of lesions. Therefore, we revised the statement in the abstract and list it as the limitation. Thank you again.

Line 38 -39: Here we show a universal algorithm can detect most types of trauma-related radiographic findings,

Line 269-271: Insufficient training data, such as hip dislocation will decrease algorithm performance. Fortunately, the conditions analyzed in this study are rarely missed by physicians.

Comment: 2. Introduction Lines 121-123

A positive point is that the investigators built off of previous work by training with weak localization labels and getting more data, using an innovative labeling approach for fast and easy localization labels (“point-based annotation”).

Response: Thank you for your constructive opinion, point-based annotation is an efficient and flexible method to provide local information.

Comment: 3. Methods

Some concern is raised regarding the discordance between the size of the test and training set (5204 PXR's versus PRX150 test set (50 hip fractures, 50 pelvic area images, and 50 normal images). The authors indicate that one of their limitations was their relatively small

training set. While it is true that more data would likely improve performance, there is no reason to believe that this is, in fact, the case (e.g., showing model performance as a function of size of the data set). Ensembling strategy both at training time (through cross fold validation) and a test time (through data augmentation) are both sound ways to improve model performance, although at a cost of inference time speed when deploying the algorithm in practice.

Response: Thank you for your inspiring opinions. The PXR 150 test set is mainly for the physician comparison test. We used 1888 PXR from 2017 as an external test set to evaluate the model performance. Following your suggestion, we performed an additional experiment to evaluate the influence of the size of the training set on the performance. The result is described in the session: Evaluation of the impact of annotation size and revised Table 4. As you mentioned, the ensembling strategy indeed increased the inference time. In our experience on the clinical testing, the model needs 5 seconds to generate the output on a GeForce GTX 1080 Ti GPU, which is tolerable in the clinical scenario.

Line 168: Evaluation of the impact of annotation size in the Results section.

Line 169- 180: The size of the annotation used for training is a key factor affecting the performance of deep learning models. To understand how PelviXNet is affected by the annotation size, we conducted experiments to train PelviXNet with 20% (N=1040), 40% (N=2081), 60% (N=3122) and 80% (N=4163) physician annotated PXR using the same settings and evaluate the trained models on the PXR2017 dataset. We also conducted another experiment to train the weakly supervised fracture detection model²² for comparison using only image-level labels automatically generated from the clinical diagnosis without manual annotation from physicians. The comparisons are summarized in Table 4. We observe that the performance of PelviXNet steadily improves (i.e., AUROC from 0.933 to 0.973) as more physician annotated PXR (from N=1040 to N=5204) are used for training. The weakly supervised model supervised by the clinical diagnosis reports an AUROC of 0.967.

Revised Table 4.

Comment:4. Methodology –

It is uncertain how the investigators aggregate labels from their three labelers, or how they chose an operating point to calculate sensitivity, specificity, and accuracy. Were dislocations definitely included in the test set?

Response: Thank you for this thoughtful comment. The three labelers labeled the images separately. When the inconsistency happens, one of the reviewers will review the clinical information and additional image exams of the patient. He will also discuss with the other two labels to make a final decision of the label. We choose the operating point based on the sensitivity greater than 90% with a tolerable specificity which is more suitable for an assisted diagnosis system. The dislocations are included in the 2017 test set which includes 1888 PXR. We only choose hip fracture and pelvic fracture for physician comparison tests because hip dislocations are seldom missed by physicians.

Line 313-316 : The three reviewers labeled the images separately. When the inconsistency happens, one of the reviewers will review the clinical information and additional image exams of the patient. He will also discuss with the other two labels to make a final decision of the label.

Comment: 5. Results Line 148 –

How was the 0.972 AUROC calculated; Normal vs. Abnormal (across all conditions)?
Mean of AUC across conditions?

Response: Thanks for your comments. The AUROC is calculated based on the Normal vs. Abnormal (across all conditions). We added a more accurate description in the sentence.

Line146 -147 : The AUROC of classifying the normal and abnormal conditions was 0.972 (95% CI, 0.960-0.983)

Comment: 6. Table 1 –

The Table 1 breakdown of demographic data shows concern of overlap of “mechanism of injury” between development and test sets, but this reviewer is uncertain if this is a typographical error.

Response: Thank you for your thoughtful comment. We did make typographical errors in Table 1. We correct them in the revised version. Thank you again.

Comment:7. Results –

No comparison is provided of the results to previous work with only image-based labels. It seems like there are more training data now as well. How do we know that the localized labelling actually helped improve results?

Response: Thanks for your constructive suggestion. We added an experiment to evaluate the impact of the amount of point annotated images to the model. The model using 80% of point annotated images outperformed the weakly supervised method using all images with image-level information only which can help the audience to understand the usefulness of localized labeling in this model. Thank you again.

Line 168 : Evaluation of the impact of annotation size in the Results section.

Line 169 -180: The size of the annotation used for training is a key factor affecting the performance of deep learning models. To understand how PelviXNet is affected by the annotation size, we conducted experiments to train PelviXNet with 20% (N=1040), 40% (N=2081), 60% (N=3122) and 80% (N=4163) physician annotated PXR's using the same settings and evaluate the trained models on the PXR2017 dataset. We also conducted another experiment to train the weakly supervised fracture detection model²² for comparison using only image-level labels

automatically generated from the clinical diagnosis without manual annotation from physicians. The comparisons are summarized in Table 4. We observe that the performance of PelviXNet steadily improves (i.e., AUROC from 0.933 to 0.973) as more physician annotated PXR (from N=1040 to N=5204) are used for training. The weakly supervised model supervised by the clinical diagnosis reports an AUROC of 0.967.

Revised Table 4.

Comment: In Table 4, model sensitivity and hip fractures in the randomly detected PXR150 is 100%, whereas in Table 2, it is only 99.4%. This doesn't seem like a random sample.

Response: Thanks for your comment. We randomly selected those images based on the clinical diagnosis before the whole analysis started. Therefore, there is probably some bias on selection. The better process is to stratify the sample according to the model performance for comparison. We will be more concerned about this problem on further study. We will list this point as a limitation in the revised manuscript. Thank you again.

Line 276-278 : In this study, we randomly selected those images based on the clinical diagnosis before the whole analysis started. Therefore, there is probably selective bias.

Comment: 8. Discussion, 199 –

The authors reference their previous work as an abstract. Has this been published?

Response: Thank you for your comment. This reference was already published in Medical Image Computing and Computer Assisted Intervention – MICCAI 2019. MICCAI 2019. Lecture Notes in Computer Science, vol 11769. Springer,

Cham. https://doi.org/10.1007/978-3-030-32226-7_51; We noted there was an error in the reference and we corrected it. thank you again.

Ref 22 : Wang Y. et al. (2019) Weakly Supervised Universal Fracture Detection in Pelvic X-Rays. In: Shen D. et al. (eds) Medical Image Computing and Computer Assisted Intervention – MICCAI 2019. MICCAI 2019. Lecture Notes in Computer Science, vol 11769. Springer, Cham

Comment 9. Figure 1 –

In Figure 1, the development data set included 5204 patients with a pelvic x-ray from a trauma center. In the annotation to the left in Figure 1, it is indicated this is labeled with “point by reviewer board with additional clinical information and advanced images.” This might potentially introduce additional bias into radiographic interpretation, relying on advanced imaging or clinical information rather than establishment of ground truth by radiographic assessment alone; i.e., “the labeler should see exactly what the model sees”.

Response: Thanks for your constructive opinion. We want to acquire the best available ground truth of fracture according to advance image or medical records. The labeler still labels the point on the fracture site which can be identified on the images, however, for the image could not be identified on PXR, the labelers would not to label it.

For example, the pelvic fracture near the sacroiliac joint is usually hard to identify by pelvic X-ray but can be detected on CT. The labeler will not put the annotation on the fracture site which can only be identified on CT.

Another example is some equivocal hip fracture can be further identified by CT or frog view. Thus, the labeler can confirm the hip fracture label using other image modalities. Therefore, the labeler could still see exactly what the model sees. Thank you again for your comment.

Reviewers' Comments:

Reviewer #1:

Remarks to the Author:

As I mentioned in the previous round of reviews, the manuscript is valuable, relevant and professionally presented. The Authors provided access to the PXR150 dataset, i.e. the dataset of 150 images which were evaluated by the physicians, and to the inference code; the code containing the network architecture and the training step was not provided and could therefore not be reviewed. Nevertheless, I could test the model with the provided database as well as with three images uploaded by myself, with results in line with those described in the manuscript. Therefore, I confirm that the model is well-performing and I can now recommend publication of the manuscript.

Reviewer #2:

Remarks to the Author:

This reviewer appreciates the additional experiments to address the concerns raised in the initial review. With the additional limitations indicated in the revised discussion, I have no additional concerns.

Response to reviewers

REVIEWERS' COMMENTS

Reviewer #1 (Remarks to the Author):

As I mentioned in the previous round of reviews, the manuscript is valuable, relevant and professionally presented. The Authors provided access to the PXR150 dataset, i.e. the dataset of 150 images which were evaluated by the physicians, and to the inference code; the code containing the network architecture and the training step was not provided and could therefore not be reviewed. Nevertheless, I could test the model with the provided database as well as with three images uploaded by myself, with results in line with those described in the manuscript. Therefore, I confirm that the model is well-performing and I can now recommend publication of the manuscript.

Fabio Galbusera

Reply: Thank you for your review and comments to increase the value of our manuscript.

Reviewer #2 (Remarks to the Author):

This reviewer appreciates the additional experiments to address the concerns raised in the initial review. With the additional limitations indicated in the revised discussion, I have no additional concerns.

Reply: Thank you for your review and comments to increase the value of our manuscript.

|